# Observation of coherent delocalized phonon-like modes in DNA under physiological conditions

Mario González-Jiménez[1], Gopakumar Ramakrishnan[1], Thomas Harwood[2], Adrian J. Lapthorn[1], Sharon M. Kelly[3], Elizabeth M. Ellis[2] & Klaas Wynne[1]

Underdamped terahertz-frequency delocalized phonon-like modes have long been suggested to play a role in the biological function of DNA. Such phonon modes involve the collective motion of many atoms and are prerequisite to understanding the molecular nature of macroscopic conformational changes and related biochemical phenomena. Initial predictions were based on simple theoretical models of DNA. However, such models do not take into account strong interactions with the surrounding water, which is likely to cause phonon modes to be heavily damped and localized. Here we apply state-of-the-art femtosecond optical Kerr effect spectroscopy, which is currently the only technique capable of taking low-frequency (GHz to THz) vibrational spectra in solution. We are able to demonstrate that phonon modes involving the hydrogen bond network between the strands exist in DNA at physiologically relevant conditions. In addition, the dynamics of the solvating water molecules is slowed down by about a factor of 20 compared with the bulk.

[1] School of Chemistry, WestCHEM, University of Glasgow, Glasgow G12 8QQ, UK. [2] Institute of Pharmacy and Biomedical Sciences, University of Strathclyde, Glasgow G4 0RE, UK. [3] Institute of Molecular Cell and Systems Biology, University of Glasgow, Glasgow G12 8QQ, UK. Correspondence and requests for materials should be addressed to K.W. (email: klaas.wynne@glasgow.ac.uk).

The processes important to the biological function of DNA (replication, transcription, denaturation and molecular intercalation) have in common that they start with the breaking of the hydrogen bonds between the bases of the nucleic acid. Driven by the torsional stress of the molecule[1], the destablization of the weak bonds leads to the splitting of a section of the double helix of DNA into single strands, forming a gap in the nucleic acid known as a transcriptional bubble[2].

The DNA molecule is not a static object, but vibrates and fluctuates on timescales from femtoseconds to nanoseconds. The origin of the rupture of the hydrogen bonds in DNA is thought to be low-frequency vibrational modes propagating along its length in the form of phonon-like modes that expand and contract the space between the bases[3–5]. Because adenine–thymine and guanine–cytosine base pairs differ in their hydrogen bonding and stacking, the physical properties of DNA that influence these waves, such as helical structure[6] or elasticity[7], depend on the local DNA sequence[8]. It has been suggested that DNA has regions where specific sequences favour the resonance between low-frequency vibrational modes[9,10], promoting the temporary splitting of a significant number of bases, even at physiological temperatures[11–13]. During the short period of time that the bubbles exist[14], the bases are exposed to the surrounding solvent, which has two effects. On one hand, bubbles expose the nucleic acid to reactions of the bases with mutagens in the environment[15], while molecular intercalators can take advantage of the gap and may insert themselves between the strands of DNA[11]. On the other hand, bubbles allow helicases access to DNA to stabilize the bubble, followed by splitting the strands to start the transcription and replication process[16]. For this reason, it is believed that DNA directs its own transcription.

Despite the importance of the low-frequency vibrational modes of DNA, evidence for their existence is only indirect[17]. The relatively weak absorption of far-infrared radiation by DNA compared with the extremely strong absorption by water has limited investigation by infrared spectroscopy[18]. However, a recent report suggested a non-thermal response of cellular gene expression to terahertz radiation[19]. The presence of low-frequency vibrational modes in DNA has been shown through Raman[20,21], Brillouin[22], inelastic X-ray[23] and inelastic neutron-scattering[24] measurements. However, all of these studies have been performed in unnatural solid DNA preparations (humidified films, fibres and so on) that modify the dynamics and are likely to introduce low-frequency lattice-phonon modes associated with crystal packing. Furthermore, inelastic neutron-scattering experiments have shown great sensitivity of the speed of sound to relative humidity. No spectroscopic experiments have been carried out under relevant physiological conditions in aqueous solution except over a very narrow frequency range $(10–35\,cm^{-1})$[25–27]. It seems highly likely that exposure of the nucleic acid to bulk liquid water will lead to strong damping of phonon-like modes, resulting in localization. Therefore, the importance of bubbles and the associated low-frequency phonon-like vibrational modes under physiological conditions remains unproven and unlikely.

In recent work, we have shown that the technique of ultrafast optical Kerr effect (OKE) spectroscopy could be used to determine the presence of delocalized phonon-like modes in a protein and prove their relevance to biological function[28]. This technique measures the low-frequency depolarized Raman spectrum in the time domain and obtains a spectrum through numerical Fourier transformation[29–31]. Spontaneous scattering techniques such as Raman spectroscopy and inelastic neutron scattering when applied to liquids and solutions become unreliable at low frequencies ($<1\,THz$) due to a very strong Rayleigh peak from elastic scattering[32]. The spectral resolution of

inelastic X-ray and neutron scattering is poor ($\sim15\,cm^{-1}$ or $0.5\,THz$), prohibiting spectral characterization especially below $\sim1\,THz$ (ref. 23). OKE spectroscopy has proven to be far superior at low frequencies, as it does not suffer from a large Rayleigh peak and its high signal to noise allows a detailed analysis of the spectra[29,30]. Furthermore, the OKE signal from liquid water is relatively weak, allowing it to be subtracted from the total signal to reveal the spectrum of the solvated biomolecule. Thus, OKE is the only technique that can reliably determine the terahertz and sub-terahertz spectra of biomolecules in physiologically relevant aqueous solution.

Here we use OKE spectroscopy to investigate the low-frequency vibrational modes of DNA to determine the presence and possible role of phonon-like modes in nucleic acids in aqueous solution. We present the OKE spectra of two DNA oligomers in phosphate buffer solution, at various degrees of denaturation as a function of increasing temperature. Two phonon-like modes associated with the hydrogen bonds of the double strand of DNA are identified. To confirm the assignments given to the observed bands, the OKE spectra of two different oligomers designed to form a double helix only when they are dissolved together are measured. Since nucleotides in solution stack even at low concentrations while not forming hydrogen bonds between them[33], the OKE spectra of cytosine are measured to investigate the influence of stacking on the nucleic acid spectra.

## Results

**OKE spectra of 20mers**. Experiments were carried out on two relatively short palindromic DNA sequences containing mostly cytosine and guanine: d(GGCGGCCCGCGCGGGCCGCC)₂ (CG 20mer), and mostly adenine and thymine: d(TTATTAAA-TATATTTAATAA)₂ (AT 20mer). OKE spectra for 10 mM solutions of the CG and AT oligomers in phosphate buffer (pH = 7) were measured at 10 K temperature intervals from 298 to 358 K. The OKE spectrum of water was subtracted at each temperature (Supplementary Note 1; Supplementary Figs 1–4; Supplementary Tables 1–4) to obtain the spectra of the solvated oligomers. At the concentration employed (0.15 M), the OKE spectrum of phosphate buffer is indistinguishable from that of water.

Figure 1 shows the experimental temperature-dependent OKE spectra of the AT 20mer and fits to theoretical expressions. Below $\sim50\,GHz$, the contribution of the molecular orientational diffusion of the oligomer is expected to dominate. Using the Stokes–Einstein–Debye equation[29], a relaxation frequency between 0.2 GHz at 298 K and 1.6 GHz at 358 K was estimated for this process (Supplementary Note 2; Supplementary Fig. 5). Because this is at significantly lower frequency than accessible in these experiments, only the high-frequency wing of the band can be seen in the spectra. Despite this, the orientational relaxation band could be fitted using a Debye function (D in Fig. 1; Methods). The relaxation time constants obtained through curve fitting are broadly consistent with the values calculated using the Stokes–Einstein–Debye equation.

The remainder of the low-frequency portion of the spectra is caused by a very broad band that can be fitted with a Cole–Cole function (Methods) with its maximum moving from 14.2 GHz at 298 K to 27.8 GHz at 358 K. This band extends to unusually high frequency and is responsible for the increase in intensity in the region between 4 and 9 THz.

The most interesting features of the OKE spectra of the AT 20mer are the changes with temperature that appear in the high-frequency part of the spectra ($>200\,GHz$). This portion of the spectrum can be fitted by four Brownian oscillators (Methods). There is only a very small change with temperature in the

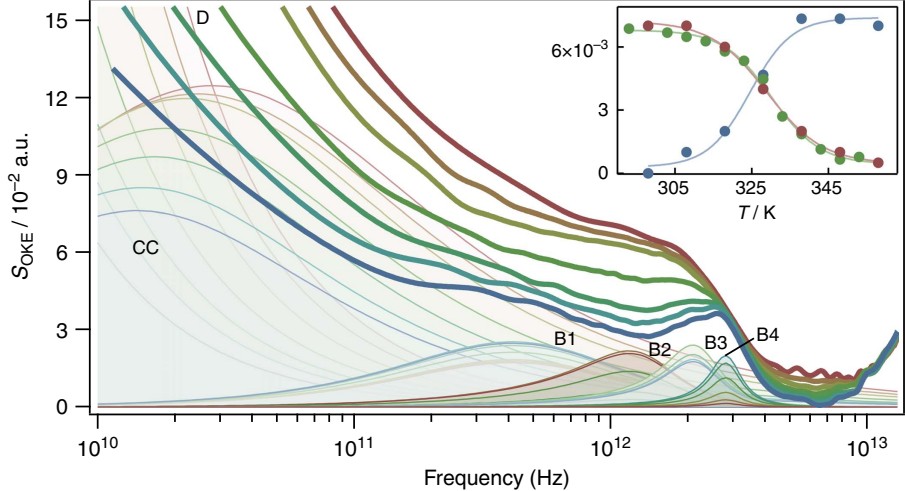

**Figure 1 | Experimental optical Kerr effect spectra and the component fit functions used to fit the data for the 20-bp AT oligomer.** The temperatures run from 298 (blue) to 358 K (red) in steps of 10 K. Each spectrum has been fitted using the combination of a Debye function (D), a Cole–Cole function (CC) and four Brownian oscillators (B1–B4). The features in the figure that change in an unexpected manner are the bands B2 and B4 (with emphasized colours). The intensity of B2 increases with temperature, while the intensity of B4 reduces. The inset shows the intensity of the bands B2 (blue) and B4 (red). They are compared with the scaled results of a circular dichroism experiment (ellipticity at 248 nm, green).

intensity of the first (B1 in Fig. 1, with the Brownian oscillator frequency $\omega_0/2\pi = 1.01$ THz and damping rate $\gamma/2\pi = 1.39$ THz) and third (B3, $\omega_0/2\pi = 2.19$ THz and $\gamma/2\pi = 0.70$ THz) bands. However, there are significant changes in the intensities of the second (B2) and fourth (B4) bands, which is unexpected. At room temperature (298 K), only B4 occurs in the spectrum but as the temperature increases, the intensity of B4 decreases, while B2 appears and increases in intensity. The inset of Fig. 1 shows the sigmoidal dependence of the intensity of these bands on temperature. A circular dichroism (CD) experiment to determine the proportion of denatured oligomer at each temperature was carried out (Supplementary Note 3; Supplementary Figs 6 and 7) showing the same sigmoidal dependence on temperature from double stranded at room temperature to single stranded at 358 K. This strongly suggests that B2 and B4 are associated with the single- and double-stranded conformation of the oligomer, respectively. The melting temperatures for the AT 20mer derived from a sigmoidal fit to the intensity of the B2 and B4 bands ($324 \pm 2$ and $329 \pm 1$ K, respectively) and from the CD experiment ($330.1 \pm 0.6$ K) are in good agreement as expected.

The remaining fit parameters of these bands are $\omega_0/2\pi = 1.38$ THz and $\gamma/2\pi = 0.75$ THz for B2, and $\omega_0/2\pi = 2.83$ THz and $\gamma/2\pi = 0.50$ THz for B4. Thus both vibrational modes have a damping rate smaller than their oscillator frequencies and are therefore underdamped. Furthermore, these bands (in particular the high frequency B4 band) cannot be fitted by a Gaussian function (Supplementary Note 4; Supplementary Fig. 8).

The temperature-dependent OKE spectra of the CG 20mer (Supplementary Note 5; Supplementary Fig. 9) show similar behaviour and can be fitted in a similar manner. However, the characteristic B4 band only starts changing at relatively high temperature consistent with results of CD, which shows that the melting temperature of the CG 20mer is higher than the accessible temperature range in the OKE experiment.

**Investigation of 13- and 17-base oligomers.** The data in Fig. 1 strongly suggest that the B2 band corresponds to single-stranded DNA and the B4 band to double-stranded DNA. To prove this, a room-temperature experiment was carried out using two oligomers of 13 and 17 bases with sequences d(CGAAAAATGTGAT) and d(CTAGATCACATTTTTCG) that minimize the possibility

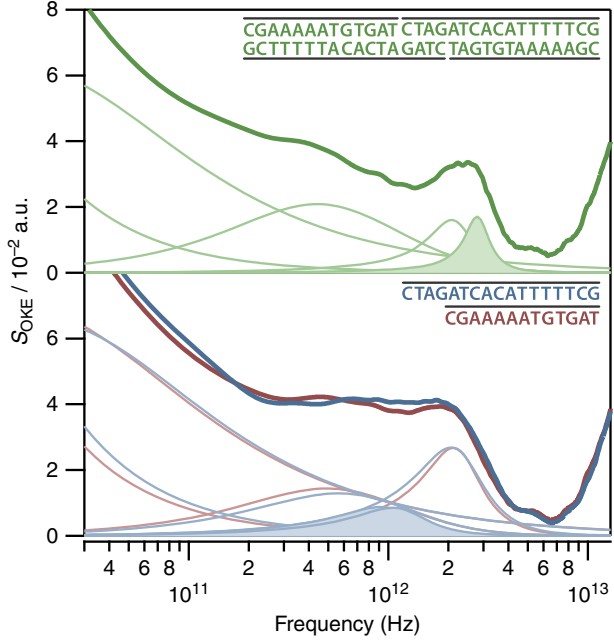

**Figure 2 | OKE spectra and fits for the solutions of the complementary oligomers of 13 and 17 bases dissolved together and separately.** Dissolved together (top) and separately (bottom, where the spectrum from the 13mer is shown in red and that of the 17mer in blue). The significant difference between top and bottom is the bands filled with colour. When the nucleic acids are single stranded, there is a band at 1.08 (13mer) and 1.20 THz (17mer). However, when they form a double strand, only a characteristic band at 2.83 THz appears.

of association between strands when they are dissolved separately. However, when they are dissolved together, the strands match perfectly to form a 30-bp double-stranded helix. OKE spectra for 10 mM solutions in phosphate buffer of each oligomer separately and both oligomer together were measured at 298 K (Fig. 2). The spectra of the oligomers separately and together can be fitted in a very similar manner. However, solutions of the 13mer and 17mer separately show unique bands at 1.08 and 1.20 THz, respectively,

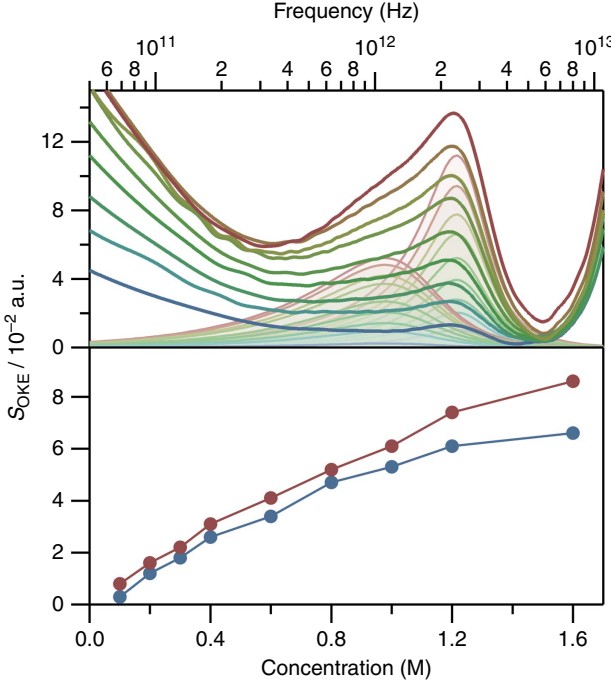

**Figure 3 | Influence of concentration on the OKE spectra of cytidine monophosphate.** The terahertz band of the spectra has been fitted using two Brownian oscillators at 1.48 and 2.58 THz (top). The effect of the concentration on the intensity of these bands (blue and red, respectively) has been plotted (bottom).

while a fourth band at higher frequency is absent. However, in the spectrum of the solution with both oligomers together, only a unique underdamped band at 2.83 THz appears (the B4 band, but not the B2 band), confirming our hypothesis.

**Base stacking in mono-nucleotide solutions.** It is now clear that the B2 and B4 bands in Fig. 1 are closely associated with the single- and double-stranded forms of DNA. The B4 band, which disappears on melting, is likely associated with hydrogen bonding between the strands. To rule out an origin in base stacking, concentration-dependent OKE experiments were carried out at room temperature on aqueous solutions of cytidine monophosphate with concentrations between 0.1 and 1.6 M. It is known that mono-nucleotides in solution tend to stack without forming intermolecular hydrogen bonds between them[33]. The OKE spectra for these solutions (Fig. 3) show a characteristic pattern with a low-frequency band (below 100 GHz) due to the orientational diffusion and a higher-frequency band between 400 GHz and 5 THz. The latter can be fitted using two Brownian oscillators, yielding the parameters $\omega_0/2\pi = 1.48$ THz and $\gamma/2\pi = 1.08$ THz, for the first oscillator, and $\omega_0/2\pi = 2.58$ THz and $\gamma/2\pi = 1.06$ THz, for the second. These vibrational modes do not coincide with the frequency of the modes observed in the oligomers and are less underdamped. The relation between intensity of the oscillators and nucleotide concentration is also plotted in Fig. 3. The curves show the same saturation behaviour that can be seen in other studies of stacking of nucleotides[33,34], that comes from the dependence on concentration of the average number of aggregated nucleotides.

**Discussion**

Thus, we have shown that the bands labelled B2 and B4 in Fig. 1 are associated with single- and double-stranded DNA respectively and that the B4 band reflects the dynamics of hydrogen bonding

between the two strands. The B4 hydrogen-bonding band can be fitted with a Brownian oscillator function, but not by a Gaussian (Supplementary Note 4). Although spontaneous Raman scattering and related techniques such as OKE cannot determine whether a band is homogeneously or inhomogeneously broadened[35–37], the good fit to a Brownian oscillator function does strongly suggest a homogeneously broadened band or at most a very narrow distribution of phonon frequencies. Therefore, this band is not a broad inhomogeneous distribution of multiple vibrational modes, instead a single delocalized normal mode of the DNA 20mer. As the frequency of this mode is high ($\omega_0/2\pi = 2.83$ THz corresponding to a period of 350 fs) and the damping rate low ($\gamma/2\pi = 0.50$ THz corresponding to a 2-ps decay), it is underdamped undergoing approximately five oscillations before vanishing.

According to the calculations using a fully atomistic model of B-DNA[4] as well as inelastic X-ray scattering experiments on humidified films of oriented DNA[23], the acoustic phonon branch for delocalized phonons in DNA peaks at ∼12 meV (3.2 THz) at the Brillouin zone edge. OKE spectroscopy has the same selection properties as Raman scattering and is therefore normally only sensitive to optical phonon modes near the zone centre where the phonon wave vector vanishes and the wavelength tends to infinity. The observed frequency of 2.83 THz is therefore consistent with an optical phonon mode with a wavelength extending throughout the entire 20mer (Brillouin zone centre). This explains the observed reduction in intensity of the B4 band with temperature, since the heating-induced denaturation disrupts the hydrogen bonds of the base pairs and thereby causes the strands to separate eliminating the phonon mode.

The phonon-like mode in double-stranded DNA observed here using OKE spectroscopy has a half-period of 177 fs, which sets the timescale for the breaking of hydrogen bonds in double-stranded DNA. However, this result is consistent with nuclear magnetic resonance experiments[2,38], fluorescence spectroscopy[39,40] and simulations[8] showing much longer timescales. These experiments measure a global conformational change in the double-stranded DNA or the probability of imino-proton exchange in an established bubble[12]. The phonon-like mode as observed here using OKE sets the approach rate in the barrier crossing that leads to bubble formation, and is therefore expected to be a much faster process.

Thus, the assumption that interaction of DNA with the surrounding water would dampen and localize the phonon mode[41] is found to be incorrect. This observation is consistent with the previous observation of similar delocalized underdamped phonon-like modes in the protein lysozyme[28].

The OKE spectrum of the AT 20mer (Fig. 1) contains a broad band peaking between 14.2 and 27.8 GHz that has been fitted to a Cole–Cole function. It is tempting to assign this CC band to some overdamped motion of the DNA insensitive to the hybridization state. However, the data analysis shows that the CC band is responsible for intensity between 4 and 9 THz, that is, above the frequency of the highest-frequency mode that can clearly be assigned to the 20mer (the B4 hydrogen-bond phonon mode). This would be unphysical (relaxational processes involving a particular molecule, by their very nature must be slower than underdamped processes of the same molecule) unless the CC band is associated with liquid water whose librational frequency is ∼25 THz. In fact, OKE and Raman scattering studies on lysozyme[28,32] observed a similar band at ∼20–40 GHz, which could be shown using neutron scattering to originate in the diffusive translational motion of water molecules in the solvation shell of the protein[32].

Thus, the CC band can similarly be interpreted as the diffusive translational dynamics of water molecules in the solvation shell of

the DNA oligomer. In bulk water at 298 K, this band peaks at[28] 245 GHz, showing a slowdown of the dynamics by a factor of 17 consistent with ultrafast solvation experiments[42]. However, this appears inconsistent with the predictions made by the jump model of water diffusion[43,44], which only predicts a slowdown by a factor 2 near a flat non-interacting surface. The surface of DNA is much more complex though with convex and concave areas. This would give rise to a greater slowdown in addition to the extreme broadening seen here[45]. At 248 K, when DNA is almost denatured, the slowdown of the dynamics is slightly larger (it changes from 17 at 298 K to 21.5 at 348 K). This is most likely caused by hydrogen bonding with the exposed bases.

Thus, the results presented here demonstrate that the inter-strand hydrogen-bond modes are coherent delocalized phonon modes even at physiological conditions. As the hydrogen bonds need to be broken for a transcription bubble to form, this result suggests that at least the initial steps of bubble formation are coherent. This is consistent with the recently observed dynamics in enzymes and enzyme–inhibitor complexes, which were similarly shown to be delocalized and coherent (although damped much more strongly than seen here in DNA). The release of water molecules from the solvation shell of DNA is thought to provide an entropic driving force for DNA–protein and anti-cancer drug binding[43]. The unexpectedly large slowdown of water dynamics by a factor of $\sim 20$ observed here will have a large effect on the dynamics and energetics of DNA binding.

## Methods

**Sample preparation.** DNA oligomers (salt-free and reverse-phase cartridge purified) were purchased from Eurofins and cytidine monophosphate from Merk. All the samples were prepared by directly dissolving in phosphate buffer (pH = 7, 0.15 M) and filtering using 0.2 μm hydrophilic polytetrafluoroethylene (PTFE) filters (Millipore) to remove dust. Before the measurements, the solutions of DNA oligomers were annealed by equilibrating them in a water bath at 363 K, followed by slow cooling down to the desired temperature. CD spectroscopy was used to determine the temperature-dependent denaturation of the DNA (Supplementary Note 3; Supplementary Figs 6 and 7).

**OKE experimental details.** The OKE data were recorded in a standard time-domain step-scan pump-probe configuration and Fourier transformed to obtain the frequency-domain reduced depolarized Raman spectrum as described previously[28–31]. A laser oscillator (Coherent Micra) provided $\sim 10$-nJ pulses (0.8 W average power) with a nominal wavelength of 800 nm at a repetition rate of 82 MHz providing a 20-fs pulse width in the sample. For the longer-timescale relaxation measurements, a second set of data was taken in a similar configuration using a higher pulse energy of typically 1 μJ provided by a regeneratively amplified laser (Coherent Legend Elite USX) at a repetition rate of 1 kHz with a pulse duration stretched to $\sim 1$ ps. Stretching the pulse enables a higher energy to be used without nonlinear effects and reduces the upper bandwidth limit allowing large step size scanning without introducing undersampling artefacts. Pump-probe OKE experiments were carried out with delay times from a few femtoseconds to a maximum of 1–4 ns, resulting in a spectral resolution (after Fourier transformation) of better than 1 GHz ($< 0.033 \, \mathrm{cm}^{-1}$)[28,29]. The sample was contained in a rectangular quartz cuvette (Starna, thickness: 1 mm) held in a brass block that was temperature controlled with a precision of $\pm 0.5$ K.

**OKE data analysis.** The OKE spectra consist of several broad overlapping bands that are analysed through curve fitting to a number of analytical functions (Supplementary Note 4). At the lowest frequencies, one finds processes associated with diffusive orientational relaxation of the molecules. The diffusive orientational relaxation of DNA in solution has been fitted here with a Debye function

$$S_D(\omega) = \mathrm{Im} \frac{A_D}{1 + i\omega\tau},$$ (1)

where $A_D$ is the amplitude of the band, $\omega$ is the angular frequency and $\tau$ is the relaxation time. The band at slightly higher frequency, which can be assigned to the diffusive relaxation of water in the solvation shell of the DNA, cannot be fitted to a Debye function due to its much greater width. This band was modelled using a Cole–Cole function

$$S_{CC}(\omega) = \mathrm{Im} \frac{A_{CC}}{1 + (i\omega\tau)^{\alpha}},$$ (2)

where $A_{CC}$ is the amplitude of the band and $\alpha$ is a parameter that accounts for the broadness of the observed band.

In the terahertz range, one finds bands from modes that are not diffusive, but critically damped or underdamped. These originate in librations, vibrations and phonon-like modes. These have been fitted using the Brownian oscillator model[28]

$$
\begin{aligned}
S_{BO}(\omega) &= \mathrm{Im} \frac{A_{BO}\omega_0^2}{\omega_0^2 - \omega(\omega + 2i\gamma)} \\
&= \mathrm{Im} \frac{A_{BO}\omega_0^2}{\omega_0^2 - \omega^2 - 2i\gamma\omega},
\end{aligned}
$$ (3)

where $\omega_0$ is the undamped oscillator angular frequency and $\gamma$ is the damping rate.

**Data availability.** The CD data and the OKE data that support the findings of this study are available in Enlighten: Research Data Repository (University of Glasgow) with the identifier http://dx.doi.org/10.5525/gla.researchdata.304.

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

## Acknowledgements

This work was supported by EPSRC grants EP/J009733/1, EP/J00975X/1, EP/K034995/1 and EP/N508792/1.

## Author contributions

All authors contributed to the study and manuscript. M.G.-J, G.R. and T.H. were responsible for data collection and numerical analysis. A.J.L., S.M.K. and E.M.E. were responsible for sample preparation and basic characterization. M.G.-J. and K.W. carried out data analysis and manuscript preparation.

## Additional information

**Competing financial interests:** The authors declare no competing financial interests.

