## [Peer review file · Nature Communications]

Reviewers' comments:

Reviewer #1 (Remarks to the Author):

Study of low frequency phonon modes in biological molecules such as DNA are important for the life science including detection of conformational changes. The authors apply femtosecond optical Kerr-effect spectroscopy to study low frequency vibrational spectra of short artificial oligomers in the broad range from GHz to several THz.

The authors claim that this spectroscopy is "the only technique that can reliably determine the spectra of biological molecules in physiologically relevant aqueous solution" in this range. The authors also claim that they identified two broad phonon-like modes associated with hydrogen bonds of DNA strands in buffer solution. However in the last decade several versions of absorption spectroscopy were used with the different degree of success as described in the book chapters [Heilweil E J, Plusquellic D F (2007) Terahertz Spectroscopy of Biomolecules. In: Dexheimer S L (ed) Terahertz Spectroscopy: Principles and Applications. Optical Science and Engineering. CRC Press, Boca Raton, pp 269-297.

T. Globus, B. Gelmont, I. Sizov. Overview of THz spectral characterization for biological identification. Chapter 10, p. 281-312, In: Biological Identification, 1st Edition. DNA Amplification and Sequencing, Optical Sensing, Lab-On-Chip and Portable Systems. Ed: Schaudies P. Woodhead Publishing, ISBN :9780857095015, 470 p., May 2014.]

As far as I know, the observation of the broad bands in the THz DNA spectra using optical Kerr- effect spectroscopy is novel and could be of interest to others in the field. This approach has been earlier applied by the authors for characterization of proteins [ref. 21 in the paper]. As a new spectroscopic method it provides information complimentary to other methods and can potentially influence the deep understanding of important phenomena related to inter-atomic vibrations in biological molecules.

This method, however, currently suffers because of the lack of spectral resolution that is required for more detail characterization. The conclusions that the authors maid regarding the observed broad modes probably related to hydrogen bonds could be much more convincing if the authors provide more information about the experimental set-up, including the sample size and shape, especially the sample thickness, cuvette size, the beam intensity, periodicity, beam energy in pulse and average, as well as the overall spectral resolution of the system. It is not clear what is the variability of observed spectra depending on modifications in experimental procedure or data manipulation [see ref. 22 in the paper]. Unfortunately, the Methods section in the current paper is very short and does not give enough information to evaluate the statements of the authors regarding the superiority of their technique. This is especially important since the authors' technique does not provide direct evidence of modes. The authors report smooth spectra, and make their conclusions based only on result of the spectra deconvolution. It seems that the spectral resolution

is not enough, which could be the result of not optimal experimental set-up, or time-domain set-up for data collection. As a minimum, the authors should add more detail information to the method section.

Reviewer #2 (Remarks to the Author):

This paper is important because it brings a crucial contribution to a subject which has been highly discussed, sometimes in harsh debates, for a long time.

There is a general agreement that DNA is a dynamical object, but the role of its fluctuations is often minimized and considered as irrelevant in a biological environment. The main argument is that, in a solvent, the dynamics is overdamped. Moreover, based on some measurements made at lower temperatures, the probability for fluctuational events physiological temperature is often considered as negligible.

The observation of these fluctuations is not trivial. As pointed out by the authors, conventional methods using Raman, Brillouin, inelastic X-ray or neutron scattering face two difficulties: i) the vibrational modes have a low frequency so that they can hardly be distinguished from the strong static scattering of the samples, ii) the need to work with oriented samples in fibers or crystals, does not allow studies in conditions close to the biological conditions.

The present work uses a technique which works with DNA solutions, and moreover allows measurements in the GHz and THz range, which is the relevant domain for the fluctuations of DNA which involve base-pair opening. This does not mean that the measurements are straightforward. The spectra (Fig. 1) contain many overlapping contributions, which must be separated and properly assigned.

The paper present a series of measurements in order to address these difficulties. First the work is performed with two DNA sequences. One of them involves the weak AT pairs, while the other one involves the strong GC pairs. This allows the authors to distinguish contributions which can be attributed to the general structure of DNA, and those which are related to the peculiarity of the base pairs. Combined with the study of the temperature dependence of the various contributions, this leads to a convincing assignment of the different component of the spectra. The fits, discussed in the supplementary materials, appear to be significant.

To reach their conclusions, the authors have also examined other measurements. Their study of short sequences, prepared either as single strands or double strands in solution allows them to confirm their assignment of the B2 (single strand contribution) and B4 (double strand contribution) that was already strongly supported by a comparison of the temperature dependence of the intensity of these modes with the thermal denaturation curve of their DNA sample. To confirm the effect of the pairing of the bases, and not of their stacking interactions, the paper also presents measurements on mono-nucleotides solutions, which are known to stack without forming hydrogen bonding.

All together the results provide a convincing evidence that the authors did observe a "delocalised

normal mode" associated to DNA pairing, and that it is not overdamped. This conclusion is strengthened by the comparison with atomistic calculations of DNA dynamics and neutron scattering measurements, which point out to the same type of optical phonon branch.

The paper goes beyond this conclusion because the authors analyze a broad band in the 15-30 GHz range, which can be assigned to the dynamics of the water molecules in the hydration shell. It suggests that the vicinity of DNA has a strong influence to slowdown the dynamics of water molecules. This is perhaps not a surprise because the important role of the interaction between water and DNA has been accepted for a long time. However, as pointed by the authors of the present paper, there may be a link between this peculiarity of surface water of DNA and the underdamped fluctuations of the base pairs. This is an interesting suggestion that could open new ways for further investigations.

In conclusion, I think that this paper addresses a very important issue, both for the physics of DNA and for its biological function, and that the authors provide convincing evidences of their statements. This is why I recommend publication.

I think however that the authors could put their result in a broader context in their presentation and discussion. The suggestion that fluctuational openings of DNA could be significant in biology is not new. This was already pointed out by von Hippel in 1965 (M.P. Printz and P.H. von Hippel, Proc. Nat. Acad. Sci. USA, 53, 363 (1965)). This seminal article should probably be cited.

This important idea had been discarded owing to the estimated low probability of the events. However a study of the temperature dependence of DNA persistence length (the original idea of von Hippel) shows that, if one examines the activation energy of the opening, at biological temperature the fluctuational events are more likely than generally assumed (Theodorakopoulos et al. PRL 108, 078104 (2012)).

Another recent experimental study also points out the importance of fluctuational opening in some DNA sequences at physiological temperature (Cuesta-Lopez, Nucleic Acid Research 39, 5276 (2011)). Although it does not deal with the dynamics of the fluctuations, this work provides an evidence supporting the main point of the present paper that fluctuational openings of DNA at physiological temperature are important, and of biological significance. The convergence of these different observations, together with references 4 and 11, already cited in the article, gives a broader impact to the present work.

Michel Peyrard,
ENS de Lyon, France

Reviewer #3 (Remarks to the Author):

In this paper, the authors proof the existence of an underdamped mode in the very low frequency part of the spectrum of a huge (huge from the perspective of vibrational spectroscopy) bio-macromolecule, DNA. I fully agree with one of the statements in the introduction of the paper ("It seems highly likely that exposure of the nucleic acid to bulk liquid water will lead to strong damping of phonon-like modes resulting in localisation.") In this regard, the experimental observation of such

a mode is indeed very interesting. I would argue that there are very few carefully done experimental papers in the literature that rigorously prove the existence of such underdamped low-frequency modes. Unfortunately, there are very many rather poor papers with wild claims of that sort in the literature, but this paper certainly sticks out in a positive sense. That is, it contains very carefully done experiments with quite a few control experiments, and the data are of extremely high quality. The rigorous experimental proof of such a mode, as contained in that paper, certainly warrants publication.

Having said that, unfortunately, many of the claims towards the end of the paper are completely speculative. In particular, I refer to the two sentences:

"Thus, the results presented here show that the DNA opening and closing processes relevant to biological function are controlled by coherent delocalised phonon modes even at physiologically relevant conditions."

and

"The degree of damping of the delocalised phonon modes in DNA has likely been optimised by evolution to maximise the efficiency of biologically relevant processes."

both of which have no justification whatsoever! To support the second sentence, one would have to modify the dephasing somehow (I wouldn't know how, but if Nature can do it, the authors should also be able to do so as well), implant the DNA in some bacteria, and then watch their performance go down.

A molecule of N atoms has $3N-6$ modes, with the density of states increasing in the low frequency part (see e.g. work by David Leitner on normal modes of proteins). A molecule of that size has 100's of modes in the relevant frequency range. A detailed assignment of vibrational spectra is already a very difficult task at high frequencies, where the density of states is comparably low, and in my opinion impossible in that low-frequency range. In simple words: By some unknown mechanism, one out of these 100 modes is underdamped. Why would that imply it is biologically relevant, and why would that mode be related to bubble formation?

In my opinion, the authors have to remove all the fancy claims from the paper, and stick to what they have: a high quality spectroscopic observation. If the paper then still is "good enough" for Nature Communication is the decision of the Editor. It certainly would be good enough for a more topical journal such as Biophys. J, J. Chem. Phys. or J. Phys. Chem.

A few more detailed comments:

- Page 4, left column: In my opinion, the distinction between a Brownian Oscillator and a Gaussian lineshape, and the conclusions on homogeneous vs. inhomogeneous broadening, is very very weak. It is by now a paradigm in the femtosecond community that linear spectra - and these are linear spectra even though they are measured by a Raman process - cannot distinguish homogeneous vs. inhomogeneous broadening, see e.g. J. Chem. Phys. 83, 2116 (1985).

- Page 4, right column top: I didn't understand how the vibrational lifetime of that mode is related to timescales observed by NMR or fluorescence. The vibrational lifetime of a mode in that frequency range will never be anywhere near 10 ns or even 50 us! But on the other hand, I also don't see why that would be important for the paper.

- Papers of the sort of Ref. [5] are among the wilder stories that exist around DNA, and I would strongly recommend the authors to motivate their work with scientifically more established approaches, which should easily be possible.

Reply to referees' comments

Reviewer #1 (Remarks to the Author):

Study of low frequency phonon modes in biological molecules such as DNA are important for the life science including detection of conformational changes. The authors apply femtosecond optical Kerr-effect spectroscopy to study low frequency vibrational spectra of short artificial oligomers in the broad range from GHz to several THz.

The authors claim that this spectroscopy is "the only technique that can reliably determine the spectra of biological molecules in physiologically relevant aqueous solution" in this range. The authors also claim that they identified two broad phonon-like modes associated with hydrogen bonds of DNA strands in buffer solution. However in the last decade several versions of absorption spectroscopy were used with the different degree of success as described in the book chapters [Heilweil E J, Plusquellic D F (2007) Terahertz Spectroscopy of Biomolecules. In: Dexheimer S L (ed) Terahertz Spectroscopy: Principles and Applications. Optical Science and Engineering. CRC Press, Boca Raton, pp 269-297.

T. Globus, B. Gelmont, I. Sizov. Overview of THz spectral characterization for biological identification. Chapter 10, p. 281-312, In: Biological Identification, 1st Edition. DNA Amplification and Sequencing, Optical Sensing, Lab-On-Chip and Portable Systems. Ed: Schaudies P. Woodhead Publishing, ISBN :9780857095015, 470 p., May 2014.]

References to the Heilweil and Globus papers were added to the introduction of the manuscript. However, the terahertz absorption spectroscopy work reported in these publications was carried out on crystalline samples or in solution but only over a narrow frequency range. A statement to this effect has been added to the manuscript.

As far as I know, the observation of the broad bands in the THz DNA spectra using optical Kerr- effect spectroscopy is novel and could be of interest to others in the field. This approach has been earlier applied by the authors for characterization of proteins [ref. 21 in the paper]. As a new spectroscopic method it provides information complimentary to other methods and can potentially influence the deep understanding of important phenomena related to inter-atomic vibrations in biological molecules.

This method, however, currently suffers because of the lack of spectral resolution that is required for more detail characterization. The conclusions that the authors made regarding the observed broad modes probably related to hydrogen bonds could be much more convincing if the authors provide more information about the experimental set-up, including the sample size and shape, especially the sample thickness, cuvette size, the beam intensity, periodicity, beam energy in pulse and average, as well as the overall spectral resolution of the system.

The Methods section has been amended to include more information about the experiment and its resolution as requested by the referee. Note that absorption of the laser beam by the sample is negligible and therefore sample pathlength is not very important. The resolution is better than 1 GHz, which corresponds to 0.033 cm^{-1} , which is now mentioned in the Methods section. Thus, the experiments do not suffer from a lack of spectral resolution but rather the vibrational/librational/diffusive bands in a condensed phase system are intrinsically broadened.

It is not clear what is the variability of observed spectra depending on modifications in experimental procedure or data manipulation [see ref. 22 in the paper]. Unfortunately, the Methods section in the current paper is very short and does not give enough information to evaluate the statements of the authors regarding the superiority of their technique. This is especially important since the authors' technique does not provide direct evidence of modes. The authors report smooth spectra, and make their conclusions based only on result of the spectra deconvolution. It seems that the spectral resolution is not enough, which could be the result of not optimal experimental set-up, or time-domain set-up for data collection. As a minimum, the authors should add more detail information to the method section.

Clearly curve-fitting is required to obtain all the different components that make up the spectra. However, the most important band (the band labelled B4 corresponding to the phonon-like band in dsDNA) can clearly be seen in the spectra at lower temperatures (see for example the blue spectrum in Figure 1) and does not require deconvolution. As discussed above, the technique does not suffer from a lack of spectral resolution, as is now discussed in the Methods section.

Reviewer #2 (Remarks to the Author):

This paper is important because it brings a crucial contribution to a subject which has been highly discussed, sometimes in harsh debates, for a long time.

There is a general agreement that DNA is a dynamical object, but the role of its fluctuations is often minimized and considered as irrelevant in a biological environment. The main argument is that, in a solvent, the dynamics is overdamped. Moreover, based on some measurements made at lower temperatures, the probability for fluctuational events physiological temperature is often considered as negligible.

The observation of these fluctuations is not trivial. As pointed out by the authors, conventional methods using Raman, Brillouin, inelastic X-ray or neutron scattering face two difficulties: i) the vibrational modes have a low frequency so that they can hardly be distinguished from the strong static scattering of the samples, ii) the need to work with oriented samples in fibers or crystals, does not allow studies in conditions close to the biological conditions.

The present work uses a technique which works with DNA solutions, and moreover allows measurements in the GHz and THz range, which is the relevant domain for the fluctuations of DNA which involve base-pair opening. This does not mean that the measurements are straightforward. The spectra (Fig. 1) contain many overlapping contributions, which must be separated and properly assigned.

The paper present a series of measurements in order to address these difficulties. First the work is performed with two DNA sequences. One of them involves the weak AT pairs, while the other one involves the strong GC pairs. This allows the authors to distinguish contributions which can be attributed to the general structure of DNA, and those which are related to the peculiarity of the base pairs. Combined with the study of the temperature dependence of the various contributions, this leads to a convincing assignment of the different component of the spectra. The fits, discussed in the supplementary materials, appear to be significant.

To reach their conclusions, the authors have also examined other measurements. Their study of short sequences, prepared either as single strands or double strands in solution allows them to confirm their assignment of the B₂ (single strand contribution) and B₄ (double strand contribution) that was already strongly supported by a comparison of the temperature dependence of the intensity of these modes with the thermal denaturation curve of their DNA sample. To confirm the effect of the pairing of the bases, and not of their stacking interactions, the paper also presents measurements on mono-nucleotides solutions, which are known to stack without forming hydrogen bonding.

All together the results provide a convincing evidence that the authors did observe a "delocalised normal mode" associated to DNA pairing, and that it is not overdamped. This conclusion is strengthened by the comparison with atomistic calculations of DNA dynamics and neutron scattering measurements, which point out to the same type of optical phonon branch.

The paper goes beyond this conclusion because the authors analyze a broad band in the 15-30 GHz range, which can be assigned to the dynamics of the water molecules in the hydration shell. It suggests that the vicinity of DNA has a strong influence to slowdown the dynamics of water molecules. This is perhaps not a surprise because the important role of the interaction between water and DNA has been accepted for a long time. However, as pointed by the authors of the present paper, there may be a link between this peculiarity of surface water of DNA and the underdamped fluctuations of the base pairs. This is an interesting suggestion that could open new ways for further investigations.

In conclusion, I think that this paper addresses a very important issue, both for the physics of DNA and for its biological function, and that the authors provide convincing evidences of their statements. This is why I recommend publication.

We are grateful for the positive comments by this referee.

I think however that the authors could put their result in a broader context in their presentation and discussion. The suggestion that fluctuational openings of DNA could be significant in biology is not new. This was already pointed out by von Hippel in 1965 (M.P. Printz and P.H. von Hippel, Proc. Nat. Acad. Sci. USA, 53, 363 (1965)). This seminal article should probably be cited.

The seminal work by von Hippel is now referred to in the first paragraph.

This important idea had been discarded owing to the estimated low probability of the events. However a study of the temperature dependence of DNA persistence length (the original idea of von Hippel) shows that, if one examines the activation energy of the opening, at biological temperature the fluctuational events are more likely than generally assumed (Theodorakopoulos et al. PRL 108, 078104 (2012)).

Another recent experimental study also points out the importance of fluctuational opening in some DNA sequences at physiological temperature (Cuesta-Lopez, Nucleic Acid Research 39, 5276 (2011)). Although it does not deal with the dynamics of the fluctuations, this work provides an evidence supporting the main point of the present paper that fluctuational openings of DNA at physiological temperature are important, and of biological significance. The convergence of these different observations, together with references 4 and 11, already cited in the article, gives a broader impact to the present work.

The important papers by Theodorakopoulos and Cuesta-Lopez are now referred to in the introduction.

Reviewer #3 (Remarks to the Author):

In this paper, the authors proof the existence of an underdamped mode in the very low frequency part of the spectrum of a huge (huge from the perspective of vibrational spectroscopy) bio-macromolecule, DNA. I fully agree with one of the statements in the introduction of the paper ("It seems highly likely that exposure of the nucleic acid to bulk liquid water will lead to strong damping of phonon-like modes resulting in localisation.") In this regard, the experimental observation of such a mode is indeed very interesting. I would argue that there are very few carefully done experimental paper in the literature that rigorously proof the existence of such underdamped low-frequency modes. Unfortunately, there are very many rather poor papers with wild claims of that sort in the literature, but this paper certainly sticks out in a positive sense. That is, it contains very carefully done experiments with quite a few control experiments, and the data are of extremely high quality. The rigorous experimental proof of such a mode, as contained in that paper, certainly warrants publication.

We are grateful for the positive comments by this referee.

Having said that, unfortunately, many of claims towards the end of the paper are completely speculative. In particular, I refer to the two sentences:

"Thus, the results presented here show that the DNA opening and closing processes relevant to biological function are controlled by coherent delocalised phonon modes even at physiologically relevant conditions."

For the double-stranded DNA to split and form a bubble, it clearly has to break the inter-strand hydrogen bonds. This is likely to be an activated process governed by an Eyring-like equation where the prefactor (the barrier approach frequency) would be limited by the frequency of the phonon-like mode. Therefore, we think that the above statement is correct although the word "show" might be too strong. We have therefore replaced it by "suggest".

and

"The degree of damping of the delocalised phonon modes in DNA has likely been optimised by evolution to maximise the efficiency of biologically relevant processes."

both of which have no justification whatsoever! To support the second sentence, one would have to modify the dephasing somehow (I wouldn't know how, but if Nature can do it, the authors should also be able to do so as well), implant the DNA in some bacteria, and then watch their performance go down.

We believe that constructing an artificial form of life that uses a novel form of genetic information carrier similar to DNA but with a greater degree of damping would be somewhat problematic... However, we agree with the referee that the use of the word "likely" is too strong and have changed the sentence accordingly.

A molecule of N atoms has $3N-6$ modes, with the density of states increasing in the low frequency part (see e.g. work by David Leitner on normal modes of proteins). A molecule of that size has 100's of modes in the relevant frequency range. A detailed assignment of vibrational spectra is already a very difficult task at high frequencies, where the density of states is comparably low, and in my opinion impossible in that low-frequency range. In simple words: By some unknown mechanism, one out of these 100 modes is underdamped. Why would that imply it is biologically relevant, and why would that mode be related to bubble formation?

The vibrational spectrum of DNA is expected to be different from that of a protein. A protein is a disordered structure (from a vibrational point of view) and one might therefore expect a large range of vibrational mode frequencies in the low-frequency part of the spectrum. However, DNA has a much greater degree of order with each base pair contributing 2 or 3 nearly-identical hydrogen bonds. A fully atomistic model of DNA (Merzel *et al.*, PRE **76**, 031917 (2007), cited in the paper) predicts that the hydrogen-bonding modes combine to form delocalised phonon modes in this quasi one-dimensional structure. These modes have strong base-pair opening character and are therefore linked to bubble formation.

In my opinion, the authors have to remove all the fancy claims from the paper, and stick to what they have: a high quality spectroscopic observation. If the paper then still is "good enough" for Nature Communication is the decision of the Editor. It certainly would be good enough for a more topical journal such as Biophys. J, J. Chem. Phys. or J. Phys. Chem.

A few more detailed comments:

- Page 4, left column: In my opinion, the distinction between a Brownian Oscillator and a Gaussian lineshape, and the conclusions on homogeneous vs. inhomogeneous broadening, is very very weak. It is by now a paradigm in the femtosecond community that linear spectra - and these are linear spectra even though they are measured by a Raman process - cannot distinguish homogeneous vs. inhomogeneous broadening, see e.g. J. Chem. Phys. 83, 2116 (1985).

It is great to see this (in)famous paper by Shaul Mukamel mentioned. This paper shows that spontaneous Raman scattering or any of the time-domain analogues such as OKE cannot effect any sort of line narrowing such as that associated with Raman photon echoes or Raman hole burning.

However, Mukamel's paper does not preclude that these techniques can distinguish the shape of spectral lines. Based on the central limit theorem it is reasonable to state that a Gaussian line shape would imply inhomogeneous broadening whereas a Lorentzian (or Brownian) lineshape would not.

- Page 4, right column top: I didn't understand how the vibrational lifetime of that mode is related to timescales observed by NMR or fluorescence. The vibrational lifetime of a mode in that frequency range will never be anywhere near 10 ns or even 50 us! But on the other hand, I also don't see why that would be important for the paper.

It appears that the manuscript was confusing at this point. Our aim was to describe the possibility of a range of timescales: from the timescale at which the barrier is approached, to the average timescale for crossing the barrier (breaking hydrogen bonds), and a timescale for the formation of large (multiple base-pair) bubbles. To avoid confusion, we have simplified the offending sentence, which we hope will make this paragraph clearer.

- Papers of the sort of Ref. [5] are among the wilder stories that exist around DNA, and I would strongly recommend the authors to motivate their work with scientifically more established approaches, which should easily be possible.

We have added a number of additional references in response to the comments made by the second referee while the original reference 5 has been removed from the introduction.

Reviewers' comments:

Reviewer #1 (Remarks to the Author):

Study of low frequency phonon modes in biological molecules such as DNA are important for the life science including detection of conformational changes. The authors apply femtosecond optical Kerr-effect spectroscopy to study low frequency vibrational spectra of short artificial oligomers in the broad range from GHz to several THz.

The observation of the broad bands in the THz DNA spectra using optical Kerr-effect spectroscopy is novel and could be of interest to others in the field. As a new spectroscopic method it provides information complimentary to other methods and can potentially influence the deep understanding of phenomena related to inter-atomic vibrations in biological molecules.

I think the revised paper is an important demonstration and analysis of low frequency phonon modes in solutions of DNA samples, and that this paper can be published in Nature Communications.

Reviewer #3 (Remarks to the Author):

The authors present a minimally revised version of their paper, and I still have my doubts. Here are my responses to the author's responses:

Original Comment: Having said that, unfortunately, many of claims towards the end of the paper are completely speculative. In particular, I refer to the two sentences:

"Thus, the results presented here show that the DNA opening and closing processes relevant to biological function are controlled by coherent delocalised phonon modes even at physiologically relevant conditions."

Author response: For the double-stranded DNA to split and form a bubble, it clearly has to break the inter-strand hydrogen bonds. This is likely to be an activated process governed by an Eyring-like equation where the prefactor (the barrier approach frequency) would be limited by the frequency of the phonon-like mode. Therefore, we think that the above statement is correct although the word 'show' might be too strong. We have therefore replaced it by 'suggest'.

Original Comment: and

"The degree of damping of the delocalised phonon modes in DNA has likely been optimised by evolution to maximise the efficiency of biologically relevant processes." both of which have no justification whatsoever! To support the second sentence, one would have to modify the dephasing somehow (I wouldn't know how, but if Nature can do it, the authors should also be able to do so as well), implant the DNA in some bacteria, and then watch their performance to go down.

Author response: We believe that constructing an artificial form of life that uses a novel form of genetic information carrier similar to DNA but with a greater degree of damping would be somewhat problematic... However, we agree with the referee that the use of the word 'likely' is too strong and have changed the sentence accordingly.

My response to the author response: Despite toning down the language, the claims are still in. In my opinion, these types of claims (which I have seen too often) hurt the paper more than they help, but this is not my decision.

Original Comment: A molecule of N atoms has $3N-6$ modes, with the density of states increasing in the low frequency part (see e.g. work by David Leitner on normal modes of proteins). A molecule of that size has 100's of modes in the relevant frequency range. A detailed assignment of vibrational spectra is already a very difficult task at high frequencies, where the density of states is comparably low, and in my opinion impossible in that low-frequency range. In simple words: By some unknown mechanism, one out of these 100 modes is underdamped. Why would that imply it is biologically relevant, and why would that mode be related to bubble formation?

Author response: The vibrational spectrum of DNA is expected to be different from that of a protein. A protein is a disordered structure (from a vibrational point of view) and one might therefore expect a large range of vibrational mode frequencies in the low-frequency part of the spectrum. However, DNA has a much greater degree of order with each base pair contributing 2 or 3 nearly-identical hydrogen bonds. A fully atomistic model of DNA (Merzel et al., PRE 76, 031917 (2007), cited in the paper) predicts that the hydrogen-bonding modes combine to form delocalised phonon modes in this quasi one-dimensional structure. These modes have strong base-pair opening character and are therefore linked to bubble formation.

My response to the author response: Ok, agreed, the spectrum of DNA will indeed be a bit simpler due to the approximate translational symmetry.

Original Comment: Page 4, left column: In my opinion, the distinction between a Brownian Oscillator and a Gaussian lineshape, and the conclusions on homogeneous vs. inhomogeneous broadening, is very weak. It is by now a paradigm in the femtosecond community that linear spectra - and these are linear spectra even though they are measured by a Raman process - cannot distinguish homogeneous vs. inhomogeneous broadening, see e.g. J. Chem. Phys. 83, 2116 (1985).

Author response: It is great to see this (in)famous paper by Shaul Mukamel mentioned. This paper shows that spontaneous Raman scattering or any of the time-domain analogues such as OKE cannot effect any sort of line narrowing such as that associated with Raman photon echoes or Raman hole burning. However, Mukamel's paper does not preclude that these techniques can distinguish the shape of spectral lines. Based on the central limit theorem it is reasonable to state that a Gaussian line shape would imply inhomogeneous broadening whereas a Lorentzian (or Brownian) lineshape would not.

My response to the author response: I very strongly disagree! First and more importantly, the multi-line fit of Fig.1 (for example) cannot distinguish between Gaussian versus Lorentzian; the difference lies in the wings of the lineshapes, but the wings are not visible due to overlap with other bands. The authors did not provide any alternative (poorer) fit by choosing a different lineshape function to make the case that one might distinguish both cases. And second, even if one could distinguish Gaussian versus Lorentzian, it is still a weak argument (despite the fact that very many people use it), as the central limit theorem very often does not hold, in particular when one is talking of hydrogen bonds. The most prominent example of a non-Gaussian but inhomogeneous OH stretch vibration of water. I realize that the authors argue that it is Lorentzian, which is indeed less model dependent. But at least in principle, there could be a non-Gaussian inhomogeneous distribution that fakes a Lorentzian line, at least to a certain extent. Again, it is a paradigm in the femtosecond community that linear spectra cannot distinguish homogeneous vs. inhomogeneous broadening, independent from any line shape arguments!

Original Comment: Page 4, right column top: I didn't understand how the vibrational lifetime of that mode is related to timescales observed by NMR or fluorescence. The vibrational lifetime of a mode in that frequency range will never be anywhere near 10 ns or even 50 us! But on the other hand, I also don't see why that would be important for the paper.

Author response: It appears that the manuscript was confusing at this point. Our aim was to describe the possibility of a range of timescales: from the timescale at which the barrier is approached, to the average timescale for crossing the barrier (breaking hydrogen bonds), and a timescale for the formation of large (multiple base-pair) bubbles. To avoid confusion, we have simplified the offending sentence, which we hope will make this paragraph clearer.

My response to the author response: I still don't understand that paragraph. If it is about an Eyring-

like equation, then what counts as an attempt-rate would be the frequency of the mode (3 THz), and not its dephasing time (which in Kramers theory would be the equivalent of friction) and even less so its vibrational relaxation time (which I would not know how it enters into Kramers theory; probably, since it is a classical theory, T2 and T1 are one-to-one related.) Again, I don't see why the vibrational relaxation time, and the fact that that it is "longer and possibly much longer" is of any relevance here.

Reply to referees' comments

Reviewer #1 (Remarks to the Author):

Study of low frequency phonon modes in biological molecules such as DNA are important for the life science including detection of conformational changes. The authors apply femtosecond optical Kerr-effect spectroscopy to study low frequency vibrational spectra of short artificial oligomers in the broad range from GHz to several THz.

The observation of the broad bands in the THz DNA spectra using optical Kerr-effect spectroscopy is novel and could be of interest to others in the field. As a new spectroscopic method it provides information complimentary to other methods and can potentially influence the deep understanding of phenomena related to inter-atomic vibrations in biological molecules.

I think the revised paper is an important demonstration and analysis of low frequency phonon modes in solutions of DNA samples, and that this paper can be published in Nature Communications.

No changes were required.

Reviewer #3 (Remarks to the Author):

The authors present a minimally revised version of their paper, and I still have my doubts. Here are my responses to the author's responses:

Original Comment: *Having said that, unfortunately, many of claims towards the end of the paper are completely speculative. In particular, I refer to the two sentences:*

"Thus, the results presented here show that the DNA opening and closing processes relevant to biological function are controlled by coherent delocalised phonon modes even at physiologically relevant conditions."

Author response: *For the double-stranded DNA to split and form a bubble, it clearly has to break the inter-strand hydrogen bonds. This is likely to be an activated process governed by an Eyring-like equation where the prefactor (the barrier approach frequency) would be limited by the frequency of the phonon-like mode. Therefore, we think that the above statement is correct although the word 'show' might be too strong. We have therefore replaced it by 'suggest'.*

We have now made a significant change to this sentence. The first replacement sentence states the experimental fact while a second sentence retains some of the initial speculation although very much toned down from the original. The new version is: *"Thus, the results presented here demonstrate that the inter-strand hydrogen-bond modes are coherent delocalised phonon modes*

even at physiological conditions. As the hydrogen bonds need to be broken for a transcription bubble to form, this result suggests that at least the initial steps of bubble formation are coherent."

Original Comment: *and*

"The degree of damping of the delocalised phonon modes in DNA has likely been optimised by evolution to maximise the efficiency of biologically relevant processes." both of which have no justification whatsoever! To support the second sentence, one would have to modify the dephasing somehow (I wouldn't know how, but if Nature can do it, the authors should also be able to do so as well), implant the DNA in some bacteria, and then watch their performance to go down.

Author response: *We believe that constructing an artificial form of life that uses a novel form of genetic information carrier similar to DNA but with a greater degree of damping would be somewhat problematic... However, we agree with the referee that the use of the word 'likely' is too strong and have changed the sentence accordingly.*

My response to the author response: *Despite toning down the language, the claims are still in. In my opinion, these types of claims (which I have seen too often) hurt the paper more than they help, but this is not my decision.*

We have now deleted the sentence "We surmise that the degree of damping of the delocalised phonon modes in DNA has been optimised by evolution to maximise the efficiency of biologically relevant processes." although we would prefer for it to stay in the manuscript.

Original Comment: *A molecule of N atoms has $3N-6$ modes, with the density of states increasing in the low frequency part (see e.g. work by David Leitner on normal modes of proteins). A molecule of that size has 100's of modes in the relevant frequency range. A detailed assignment of vibrational spectra is already a very difficult task at high frequencies, where the density of states is comparably low, and in my opinion impossible in that low-frequency range. In simple words: By some unknown mechanism, one out of these 100 modes is underdamped. Why would that imply it is biologically relevant, and why would that mode be related to bubble formation?*

Author response: *The vibrational spectrum of DNA is expected to be different from that of a protein. A protein is a disordered structure (from a vibrational point of view) and one might therefore expect a large range of vibrational mode frequencies in the low-frequency part of the spectrum. However, DNA has a much greater degree of order with each base pair contributing 2 or 3 nearly-identical hydrogen bonds. A fully atomistic model of DNA (Merzel et al., PRE 76, 031917 (2007), cited in the paper) predicts that the hydrogen-bonding modes combine to form delocalised phonon modes in this quasi one-dimensional structure. These modes have strong base-pair opening character and are therefore linked to bubble formation.*

My response to the author response: *Ok, agreed, the spectrum of DNA will indeed be a bit simpler due to the approximate translational symmetry.*

OK.

Original Comment: Page 4, left column: *In my opinion, the distinction between a Brownian Oscillator and a Gaussian lineshape, and the conclusions on homogeneous vs. inhomogeneous broadening, is very very weak. It is by now a paradigm in the femtosecond community that linear spectra - and these are linear spectra even though they are measured by a Raman process - cannot distinguish homogeneous vs. inhomogeneous broadening, see e.g. J. Chem. Phys. 83, 2116 (1985).*

Author response: *It is great to see this (in)famous paper by Shaul Mukamel mentioned. This paper shows that spontaneous Raman scattering or any of the time-domain analogues such as OKE cannot effect any sort of line narrowing such as that associated with Raman photon echoes or Raman hole burning. However, Mukamel's paper does not preclude that these techniques can distinguish the shape of spectral lines. Based on the central limit theorem it is reasonable to state that a Gaussian line shape would imply inhomogeneous broadening whereas a Lorentzian (or Brownian) lineshape would not.*

My response to the author response: *I very strongly disagree! First and more importantly, the multi-line fit of Fig.1 (for example) cannot distinguish between Gaussian versus Lorentzian; the difference lies in the wings of the lineshapes, but the wings are not visible due to overlap with other bands. The authors did not provide any alternative (poorer) fit by choosing a different lineshape function to make the case that one might distinguish both cases. And second, even if one could distinguish Gaussian versus Lorentzian, it is still a weak argument (despite the fact that very many people use it), as the central limit theorem very often does not hold, in particular when one is talking of hydrogen bonds. The most prominent example of a non-Gaussian but inhomogeneous OH stretch vibration of water. I realize that the authors argue that it is Lorentzian, which is indeed less model dependent. But at least in principle, there could be a non-Gaussian inhomogeneous distribution that fakes a Lorentzian line, at least to a certain extend. Again, it is a paradigm in the femtosecond community that linear spectra cannot distinguish homogeneous vs. inhomogeneous broadening, independent from any line shape arguments!*

As suggested by the editor, we have significantly extended “Supplementary note 4 – Nonlinear curve-fitting procedures” with a paragraph describing fits of the data with anti-symmetrised Gaussian and Brownian oscillator functions (data and fits shown in Supplementary Figure 8). We now show that the fits with Brownian oscillator functions are much better as judged by the respective χ^2 values. We have added a brief discussion on homogeneous vs. inhomogeneous broadening also discussing the caveat presented by the 1985 Mukamel paper. The Supplementary note is referred to in the main text at the appropriate point in the Results section. In the Discussion section, we have added the sentence “*Although spontaneous Raman scattering and related techniques such as OKE cannot determine whether a band is homogeneously or inhomogeneously broadened, the good fit to a Brownian oscillator function does strongly suggest a homogeneously broadened band or at most a very narrow distribution of phonon frequencies.*” This should fully address the referee’s concerns while not changing any of the conclusions.

Original Comment: Page 4, right column top: I didn't understand how the vibrational lifetime of that mode is related to timescales observed by NMR or fluorescence. The vibrational lifetime of a mode in that frequency range will never be anywhere near 10 ns or even 50 us! But on the other hand, I also don't see why that would be important for the paper.

Author response: It appears that the manuscript was confusing at this point. Our aim was to describe the possibility of a range of timescales: from the timescale at which the barrier is approached, to the average timescale for crossing the barrier (breaking hydrogen bonds), and a timescale for the formation of large (multiple base-pair) bubbles. To avoid confusion, we have simplified the offending sentence, which we hope will make this paragraph clearer.

My response to the author response: I still don't understand that paragraph. If it is about an Eyring-like equation, then what counts as an attempt-rate would be the frequency of the mode (3 THz), and not its dephasing time (which in Kramers theory would be the equivalent of friction) and even less so its vibrational relaxation time (which I would not know how it enters into Kramers theory; probably, since it is a classical theory, T₂ and T₁ are one-to-one related.) Again, I don't see why the vibrational relaxation time, and the fact that that it is "longer and possibly much longer" is of any relevance here.

We apologise for misunderstanding this comment by the referee the first time! Their comment is absolutely correct: the relevant timescale is not the dephasing time but the phonon (vibrational) half period. Thus, we have deleted two short sentences on the dephasing time of the phonon-like mode and replaced it by: *"The phonon-like mode in dsDNA observed here using OKE spectroscopy has a half-period of 177 fs, which sets the timescale for the breaking of hydrogen bonds in dsDNA."*

Reviewers' Comments:

Reviewer #3 (Remarks to the Author)

the authors have adequately addressed my concerns, and the paper now can be published, in my opinion